Antimicrobial and scavenging potential of green synthesized silver/manganese bimetallic nanoparticles using Euphorbia cactus extract

Al-Hamoud Gadah A. 1
Amina Musarat mamina@KSU.EDU.SA 1
Al-Musayeib Nawal M. 1
Alhabardi Samiah 2
Haq Mohsin Ul 3
Akhtar Saeed sakhtar@inaya.edu.sa 3
1 Department of Pharmacognosy, College of Pharmacy, King Saud University , Riyadh , Saudi Arabia
2 Pharmaceutics Department, College of Pharmacy, King Saud University , Riyadh , Saudi Arabia
3 College of Applied Medical Sciences, Inaya Medical College , Riyadh , Saudi Arabia
Sotelo-Mundo Rogerio
Electronic publication date: 2025 Oct 27
Publication date: 2025
Volume: 13
Electronic Location ID: e20244
Received 2025 Feb 27; Accepted 2025 Sep 25
Copyright: ©2025 Al-Hamoud et al.
Copyright year: 2025
Copyright holder: Al-Hamoud et al.
License: This is an open access article distributed under the terms of the Creative Commons Attribution License, which permits unrestricted use, distribution, reproduction and adaptation in any medium and for any purpose provided that it is properly attributed. For attribution, the original author(s), title, publication source (PeerJ) and either DOI or URL of the article must be cited.
License URL: https://creativecommons.org/licenses/by/4.0/

Keywords: Bimetallic, Silver, Manganese, Euphorbia cactus, Antimicrobial, Antioxidant

Funding: Ongoing Researchers Funding Program ORF-2025-878 King Saud University, Riyadh, Saudi Arabia This research project was supported by the Ongoing Researchers Funding Program, (ORF-2025-878), King Saud University, Riyadh, Saudi Arabia. The funders had no role in study design, data collection and analysis, decision to publish, or preparation of the manuscript.

==============================
The fabrication of novel nanomedicines in the pursuit of alternative therapeutics has increasingly embraced eco-friendly strategies. This study reports the phytogenic synthesis of bimetallic silver-manganese nanoparticles (Ag/MnNPs) through bio-reduction using non-toxic extract from aerial part of Euphorbia cactus as a sustainable and environmentally benign reducing and stabilizing agent. Advanced spectroscopic and microscopic characterization techniques were applied to elucidate the physico-chemical features of green synthesized Euphorbia cactus-mediated Ag/MnNPs (EC-Ag/MnNPs). The formed EC-Ag/MnNPs were predominantly spherical and uniform, with an average size of 18.32 nm. Fourier transform infrared spectroscopy (FTIR) analysis revealed the existence of multiple functional groups, indicating the involvement of Euphorbia cactus phytoconstituents in the synthesis, reduction and stabilization of EC-Ag/MnNPS. The energy-dispersive X-ray (EDX) estimation confirmed the elemental composition, showing 43.62% of silver and 18.71% manganese content. Both biogenic bimetallic EC-Ag/MnNPs and Euphorbia cactus (EC) extract exhibited excellent antimicrobial and scavenging potential. The bimetallic EC-Ag/MnNPs exerted superior antibacterial efficacy, with maximum inhibition zones of 38.15 ± 0.32 mm against Escherichia coli and 36.81 ± 0.51 mm against Salmonella Typhi. EC-Ag/MnNPs also showed remarkable antifungal efficacy against Candida glabrata (35.10 ± 0.39 mm) and Candida parapsilosis (33.82 ± 0.97 mm). Additionally, the biosynthesized EC-Ag/MnNPs showed strong, dose-dependent antioxidant activity, achieving a maximum scavenging efficiency of 96.12% at a concentration of 80 µg/mL. The results demonstrated significant antimicrobial and antioxidant properties of green-synthesized EC-Ag/MnNPs, underscoring their potential application in antimicrobial formulations and enhancement of medical device functionality.

Introduction

Overuse of antibiotics has led to the alarming rise in bacterial resistance. Global epidemics caused by different microbes currently represent a major risk to public safety and economic development of countries (Bloom & Cadarette, 2019; Baker et al., 2022). Research indicates that bacterial resistance is a significant barrier in combating pathogenic microbes (Salam et al., 2023). In 2019, antibiotic-resistant diseases were responsible for 1.27 million direct deaths and contributed to an additional 4.95 million indirect fatalities, surpassing the death tolls of malaria and AIDS (Coque et al., 2023). If this trend persists, antibiotic-resistance infections could cause 10 million deaths annually by 2050 (Brussow, 2024). Thus, the development of novel and effective antibacterial substances is of utmost importance.

Nanoparticles effectively conquer bacterial resistance effectively by directly killing, membrane obstruction, and targeting specific mechanisms (Xie et al., 2023). The inorganic metal nanomaterials are less intricate, less hazardous, and less prone to develop resistance than their organic counterparts. Their efficacy for controlling microbes is attributed to their small size, large surface area, and surface charge. These attributes along with their excellent scalability, stability, and biocompatibility, make them extremely intriguing candidates for medicinal and environmental applications (Kumar et al., 2021). Moreover, resistance to nanoparticles often arises from a combination of genetic mutations, underlining their potential to fight against multiple-drug-resistant-pathogens (Prasad et al., 2021). Among different inorganic metal nanomaterials, silver nanoparticles (AgNPs) have been extensively researched owing to their remarkable antibacterial efficacy and low susceptibility to resistance development (More et al., 2023). They exhibit antimicrobial properties through the destruction of bacterial membranes, the production of reactive oxygen species (ROS) that enhance microbial inhibition, and the release of silver (Ag+) ions that disrupt DNA and protein functions (Chmagh et al., 2024). Bondarenko et al. (2018) further showed that AgNPs may specifically target the inner membranes of bacteria, providing a better understanding of their mechanism of action. Concurrently, manganese nanoparticles (MnNPs) received significant interest for their distinct physicochemical features and size-dependent catalytic conduct (Haque, Tripathy & Patra, 2021). They had a variety of biological effects, including antioxidant, antibacterial, antifungal, and anticancer properties (Hoseinpour & Ghaemi, 2018). Numerous studies have shown that MnNPs exhibit potent antibacterial effects, mainly through processes involving the generation of ROS, the release of metal ions, and the internalisation of nanoparticle (Amatya & Shrestha, 2021; Patil et al., 2024). Despite having antibacterial capability similar to that of AgNPs, MnNPs are not as stable due to issues like oxidation during synthesis and storage. Therefore, even though metal-based nanoparticles appear to have promising biofunctionalities, concern related to toxicity, cellular absorption, and chemical stability often limit their usage (Franco et al., 2022). A conceivable approach to overcome such limitations is the fabrication of bimetallic nanoparticles that allow metals to interact synergistically to improve their structural integrity, physicochemical characteristics, and overall biological performance (Dlamini, Basson & Pullabhotla, 2023).

Bimetallic nanoparticles (BNPs) gained more attention lately than their monometallic counterparts due to their distinct optical, magnetic, electrical, thermal, plasmonic, catalytic and biological properties (Srinoi et al., 2018). These enhanced qualities arise from their larger surface area and the synergistic interactions among the constituent metals, thereby rendering them ideal for use in drug delivery, imaging, sensing, catalysis, and ecological restoration (Sudhakar et al., 2024; Medina-Cruz et al., 2020). Incorporation of manganese (Mn) with noble metals like silver or gold in biologically produced BNPs often results in enhanced performance due to combinatorial synergies (Zhang et al., 2023; Elbasuney et al., 2024; Gorlin et al., 2014). This interaction offers them unique morphological, catalytic and biological attributes in contrast to their monometallic analogues. The biosynthesis of bimetallic silver-manganese nanoparticles (Ag/MnNPs) utilizing extracts of different plant species has been explored, producing nanoparticles with sustained antibacterial efficacy and low genotoxicity (Anguraj et al., 2023). For instance, Ag/MnNPs prepared from leaf extract of Arachis Pintoi and Cordia macleodii has exhibited potent antibacterial action against both Gram-negative and Gram-positive bacteria (Tien et al., 2020; Deshpande, Gothalwal & Chandra, 2023). Similarly, Ag-doped manganese dioxide (MnO2) NPs derived from Cucurbita pepo leaf extract haven shown effectiveness towards food- and water-borne microorganisms (Krishnaraj et al., 2016). In addition, manganese oxide-graphene oxide-silver (MnO-GO-Ag) nanocomposite produced from Fagonia arabica extract demonstrated strong antioxidant and anti-inflammatory properties (Mansoor et al., 2022), while Ag-MnO2 nanoparticles made from Chelidonium majus and Vinca minor exhibited potent cytotoxic effects on A375 and HaCaT cells (Ciorita et al., 2020). These combined Ag/MnNPs effects unveil unique features not found in their respective monometallic forms, underscoring the significance of further investigation on their potential for a variety of biomedical and environmental applications.

Bimetallic nanoparticles are created by combining two distinct metal elements, resulting in diverse structural configurations and morphologies (Idris & Roy, 2023). Green synthesis is an intriguing approach for nanoparticle synthesis, has received recognition for its cost effectiveness, operational simplicity, eco-friendliness, improved stability, and rapid processing (Jain et al., 2024). The biogenic preparation of bimetallic nanoparticles involves combination of two distinct aqueous metal solutions with a green reducing agent, like a plant extract (Larranaga-Tapia et al., 2024). The plant extracts known as ‘blend of phytoconstituents’ such as polyphenols, aldehydes, ketones, proteins and amides, acts as both a reducers and stabilizers in the nanoparticle biosynthesis (Scala et al., 2022). The phytoconstituents are adhered to the nanoparticles surface via dangling bonds enhancing their biocompatibility and making them more suitable for use in biomedical applications. The size and form of manufactured nanoparticles can be significantly altered by phytochemical extracts made from various plant tissues and solvents (Filho et al., 2023). Moreover, the inherent toxicological properties of the biological extracts employed in the manufacture of nanoparticle formulations can affect their toxicity (Kulkarni et al., 2023). Thus, using a non-toxic reducing agent that is biologically generated is necessary for accurately evaluating the biological characteristics of the produced nanoparticles.

Euphorbia cactus (Family: Euphorbiaceae), a miniature tree or spiny shrub, was selected as botanical material for this study due to its numerous uses in traditional medicine and varied biological activities such as treatment for bronchitis, asthma, chest congest, as well as its genotoxic, antitumor, and antiangiogenic properties (Al-Hamoud et al., 2022). The extracts of Euphorbia cactus are rich in diverse secondary metabolites such as triterpenes, flavonoids, phenolics, coumarin, amino acids, proteins, enzymes, and carbohydrates (Aati et al., 2022). These bioactive compounds make Euphorbia cactus an excellent candidate for green synthesis, as they facilitate in the bio-reduction, capping, and stabilization of bimetallic nanoparticles (BNPs). To the best of our knowledge, however, Euphorbia cactus extracts have not yet been employed for the synthesis of bimetallic green synthesized Euphorbia cactus-mediated Ag/MnNPs (EC-Ag/MnNPs).

In the current study, we aimed to establish an eco-friendly and sustainable method for preparing bimetallic EC-Ag/MnNPs using a biodegradable, non-toxic aqueous extract derived from the aerial parts of Euphorbia cactus as a reducing agent in the redox-mediated formation of the nanostructure. This green approach addresses the increasing demand for environmentally responsible nanoparticle synthesis techniques. The formed EC-Ag/MnNPs nanostructures were comprehensively characterized, encompassing spectroscopic, morphological, and elemental analyses. Furthermore, the fabricated bimetallic EC-Ag/MnNPs were explored for their free radical scavenging against DPPH assay and antimicrobial potential towards panel of human pathogens using the conventional agar well diffusion technique.

Materials and Methods

Chemicals

Silver nitrate (AgNO3), manganese chloride tetrahydrate (MnCl2. 4H2O), nutrient agar, Mueller Hinton agar, dextrose agar was acquired from Sigma-Aldrich and utilized exactly as supplied.

Botanical material and extraction

Aerial parts of Euphorbia cactus were gathered from the southern region of Saudi Arabia (coordinates 17°15′01.2″N, 43°06′40.6″E) in November 2019. The botanical material was taxonomically identified and authenticated by Dr. Ali Mohammed Alzahrani. A voucher specimen was assigned identification number (EC-14984) and placed in the herbarium of Pharmacognosy Department, College of Pharmacy (King Saud University) to ensure the reproducibility of the study. The collected botanical material was initially washed with tap water to eliminate dirt and unwanted foreign matter, followed by rinsing with double-distilled water. It was then shade-dried for a week at ambient temperature. After drying, the material was chopped and coarsely powdered. For extract preparation, 10 g of finely powdered Euphorbia cactus aerial parts were refluxed in a 250 mL round-bottom flask with 150 mL of double-distilled deionised water at 70 °C for 1 h. After cooling to ambient temperature, the extract was filtered through Whatman No. 1 filter paper, centrifuged for 5 min at 6,000 rpm and the resulting supernatant was retained at 4 °C for experimental purposes.

Phytochemical screening

The crude extract of Euphorbia cactus aerial parts was qualitatively analyzed for the existence of different phytochemicals, including alkaloids, phenolic compounds, flavonoids, terpenoids, tannins, and saponins, reducing sugars and glycosides using standard procedures described by Adil et al. (2024)

Biogenic synthesis of bimetallic EC-Ag/MnNPs

Biometatllic EC-Ag/MnNPs were biosynthesized using silver nitrate and manganese chloride tetrahydrate as precursor salts, with an extract from the aerial parts of Euphorbia cactus serving as the reducing agent. Briefly, two 100 mL stock solutions of 0.1 M AgNO3 (1.69 mg) and 0.1 M MnCl2 ⋅ 4H2O (1.60 mg) were prepared in double-distilled water. Equal volumes of 60 mL each precursor solution was combined and heated at 80 °C for 20 min with constant stirring at 4,000 rpm using a magnetic stirrer. Then, 30 mL of freshly prepared Euphorbia cactus plant extract (2 mg/mL) was poured into the reaction mixture, and stirred continuously for 6 h at 40 °C. Subsequently, pH of solution was adjusted to 10 by the gradual addition of 0.2 M sodium hydroxide (NaOH), resulting in a noticeable color shift from brown to dark brown and eventually to black, signifying the formation of EC-Ag/MnNPs (Nyabadza et al., 2023). Unreated plant constituents and residual metal precursors in the reaction mixture were eliminated by centrifugation for 10 min at 10,000 rpm. The nanoparticles obtained were washed thoroughly with double-distilled water, rinsed with ethanol, then air-dried at 80 °C in a hot air oven and stored at 40 °C for later use.

Characterization

The biogenic synthesis of EC-Ag/MnNPs was characterized by different spectral and microscopic techniques. Preliminary optical characterization was conducted by recording the ultraviolet-visible (UV-Vis) absorption spectrum in the range of 300–600 nm using a UV-1800 spectrophotometer (Shimadzu, Kyoto, Japan) at a scanning speed of 1,000 nm/min. Tauc’s plot was employed to estimate the optical bandgap. The structural properties, size and phase purity of the biosynthesized bimetallic EC-Ag/MnNPs were examined through X-ray powder diffraction (XRD) analysis using a Rigaku Ultima IV diffractometer (Rigaku, Tokyo, Japan) equipped with a CuKα radiation source (λ = 1.540 Å, 30 kV, 30 mA), and the data was recorded over a 2θ range of 20–80°. Fourier transform infrared spectroscopy (FTIR) analysis was conducted on a Shimadzu IR Affinity-1 (Shimadzu, Kyoto, Japan) within the 400–4,000 cm−1 range to detect functional moieties in the pre-synthesized EC-Ag/MnNPs and plant extract. Both the fabricated EC-Ag/MnNPs and plant extract were diluted in KBr and analyzed using the attenuated total reflectance (ATR) technique. The surface morphology, average particle size and elemental content of the formed bimetallic EC-Ag/MnNPs were examined using a scanning electron microscope (SEM, JEOL JSM-6390) operated at 15 kV accelerating voltage and coupled with energy-dispersive X-ray spectroscopy (EDX). EDX was employed to quantify the elemental distribution in terms of weight and atomic percentages. The morphology and structural pattern of EC-Ag/MnNPs were confirmed by transmission electron microscopy (TEM) and selected area electron diffraction (SAED) (JEOL-2100; JEOL Ltd., Tokyo, Japan).

Antimicrobial activity

Media preparation

The nutrient agar media was developed by dissolving 28 g of powder nutrient agar (Sigma-Aldrich, Darmstadt, Germany) in one L of distilled water and then brought to boil. The solution was autoclaved for 15 min at 121 °C and subsequently cooled to ambient temperature. After cooling, agar was transferred into Petri plates and left undisturbed for 30 to 40 min to solidify completely.

Microbial strains

A total of eight clinical isolates were tested for antimicrobial efficacy of EC-extract and biosynthesized EC-Ag/MnNPs. The bactericidal activity was assessed against six bacterial strains, comprising of three Gram positive bacteria-Staphylococcus aureus, Staphylococcus pneumonia, and Methicillin-resistant Staphylococcus aureus (MRSA); and three Gram negative bacteria-Enterobacter cloacae, Escherichia coli, and Salmonella Typhi. Antifungal activity was tested against Candida glabrata and Candida parapsilosis. Prior to testing, all selected strains of bacteria and fungi were revived on Mueller-Hinton Agar (MHA) and Dextrose Agar (SDA) plates, respectively.

Antimicrobial activity assay

The antibacterial effect of EC-extract and biogenic bimetallic EC-Ag/MnNPs was assessed by well diffusion method (Burygin et al., 2009). Overnight-grown bacterial cultures in broth were adjusted to a final inoculum density of 100 µL: 0.1A600, or around 3.2 × 108 CFU/mL. A sterile cotton swab was employed to evenly spread 20 µL of the bacterial suspension onto the surface of a sterile agar plate (20 mL) and allowed to dry for about 3 min. Test solutions of EC-extract and EC-Ag/MnNPs were aseptically prepared in distilled water and applied individually. Varied concentrations (25, 50, and 100 µg/mL) of test solution were dispensed into six mm diameter sterile well and three replicates were prepared for each sample. Commercially available Gentamicin (10 µg on each six mm disks) was employed as the positive control, while untreated bacterial/fungal cultures served as the negative control. These plates were then kept for an incubation period of 24 h at 37 °C. Antibacterial efficacy was ascertained by measuring the diameter (in mm) of the inhibition zones surrounding the wells using a transparent ruler. Three evaluations of each test solution were conducted, and the mean outcomes were presented.

Estimation of minimum inhibitory concentration

The broth microdilution procedure was applied to establish the minimum inhibitory concentration (MIC). Test samples were serially two-fold diluted in Mueller Hinton broth (MHB) within 96-well plates to accomplish final concentrations ranging from 0.192 mg/mL to 100 mg/mL for the EC-extract, and from 0.095 mg/mL to 50 mg/mL for the EC-Ag/MnNPs. Each well was inoculated with 100 mL log-phase bacterial culture adjusted to approximately 106 CFU/mL. Following 16–18 h of incubation at 37 °C, the microbial growth was accessed by monitoring turbidity. The MIC was defined as the lowest dose of test sample that visibly impeded microbial growth, indicated by a clear, non-turbid well. Three runs of each test were conducted.

Estimation of minimum bactericidal concentrations

The wells exhibiting the bacterial inhibition, as determined by the broth microdilution technique, were plated on MHA, while those showing fungal inhibitions were plated onto dextrose agar. The plates were then incubated at 37 °C for 16–18 h. The minimum bactericidal concentrations (MBCs) was defined as the lowest dose of extract or nanoparticles that resulted in 99.9% microbial killing, indicated by the absence of visible growth on the agar plates.

Morphological alteration in most sensitive bacterial strains

The two most sensitive bacterial strains, Escherichia coli and Salmonella Typhi, were exposed to EC-extract and bimetallic EC-Ag/MnNPs for 24 h. Morphological alterations in the treated Escherichia coli and Salmonella Typhi cells were compared with their respective untreated control cells. Following incubation, the bacterial cells were centrifuged for 10 min at 5,000 rpm, and the resulting pellets were rinsed with sterile PBS. The samples were then fixed at 4 °C for 4 h using a fixative solution comprising of 2% paraformaldehyde and 2.5% glutaraldehyde with intermittent vortexing. Finally, the fixed cell biomass was mounted onto glass cover slips, and morphological alterations in the cell wall were monitored under SEM.

Antioxidant activity

The scavenging attributes of biosynthesised bimetallic EC-Ag/MnNPs, aqueous EC-extract, and standard L-ascorbic acid against the free radical DPPH was evaluated at varying concentrations (20–80 µg/mL) following a slightly modified procedure of Erenler et al. (2021). A two mL aliquot of DPPH methanol solution was momentarily combined with 0.5 mL of each test sample solution and L-ascorbic acid at the respective concentrations. The mixtures were thoroughly vortexed and rested for about 30 min at ambient temperature in the dark without any disruptions. Following incubation, the absorbance was recorded at 517 nm using a spectrophotometer. All measurements were taken three times. A decrease in absorbance signified a stronger free-radical scavenging effect. The scavenging capacity of each test sample was computed using following formula Radical scavenging activity%=100−AC−ASAC×100

where Ac stands for the control absorbance and As for the sample absorbance. The outcome was analyzed using linear regression mode and the results were graphically presented as mean ± standard deviation.

Statistical analysis

The data were analyzed using MS Excel 2007 and expressed as mean ± standard deviation (SD) from three independent replicates. A statistical difference among groups was evaluated using one-way analysis of variance (ANOVA), followed by Tukey’s post hoc test, and performed with the trial version of StatPlus 2009 Professional. A p-value <0.05 was regarded as statistically significant.

Results and Discussion

Phytochemical characterization of prepared extract

The diverse phytochemicals identified in Euphorbia cactus showcase its potential as a rich source of bioactive compounds, as summarized in Table 1. The aerial portions of plants contain various fascinating groups of secondary metabolites, including triterpenes, flavonoids, steroids, coumarin, and phenolics (Aati et al., 2022). Terpenoids, the most abundant constituents in this species, exhibit variety of pharmacological attributes that underpin their extensive therapeutic uses (Kgosiemang et al., 2025; Al-Hamoud et al., 2022). These bioactive compounds render Euphorbia cactus an ideal candidate for green synthesis by enabling the bioreduction, capping, and stabilization of bimetallic nanoparticles (BNPs).

Table 1 Phytochemical screening of aqueous extract of Euphorbia cactus aerial parts.

Phytocemical	Reagent	Status*	
Triterpenes	Salkowski test	+++	
Tannins	Foam test	++	
Flavonoids	Shinoda test	+++	
Phenols	Ferric chloride	++	
Steroids	Ferric chloride	++	
Coumarins	NaOH test	+	
Alkaloids	Wagner’s	++	
Cardiac glycosides	Keller Killian’s	++	
Saponins	Foam test	–	
Reducing sugars	Benedict’s test	+	
Proteins	Biuret test	+	
Notes.

* (+++), significant amount; (++), in appreciable amount; (–), absent.

Biosynthesis of bimetallic EC-Ag/MnNPs

A green phytogenic approach was employed for the fabrication of EC-Ag/MnNPs using an aqueous extract from aerial parts of Euphorbia cactus as natural reducing agents. This eco-friendly approach offers several advantages including, simplicity, environmentally benign, free from toxic chemicals, high-energy inputs, reducing or capping agents and protracted treatments. By optimizing various parameters in the reaction solution, the nanoparticles were synthesized rapidly with high stability. The initial visual confirmation of bimetallic EC-Ag/MnNPs biosynthesis was a distinct colour transition in the extract from reddish-brown to yellow. The reaction medium was water, and the fabrication was conducted at relatively at low temperatures (between 30 and 60 °C), followed by calcination of the nanoparticles at 400 °C. The silver nitrate and manganese chloride were mixed with Euphorbia cactus, allowing silver ions from AgNO3 and manganese ions from MnCl2 to bind to plant proteins and water-soluble components through –OH and –COOH groups. This interaction induced conformational changes in the protein molecules, facilitating the transformation of the captured metal ions into EC-Ag/Mn nanoparticles (Kazemi et al., 2023). In general, the synthesis of metal nanoparticles using plants or plant extracts involves three main steps: (1) activation phase—metal ions are reduced, initiating the nucleation of metal atoms; (2) growth phase—nanoparticles increase in size as their thermodynamic stability improves, with smaller particles coalescing into larger ones through mechanisms such as heterogeneous nucleation and Ostwald ripening; (3) termination phase—the final size and shape of the nanoparticles are determined. The formed bimetallic EC-Ag/MnNPs were established by various spectroscopic (UV-vis, XRD, FTIR) and microscopic (SEM, EDX, TEM).

Structural characterization of bimetallic EC-Ag/MnNPs

The optical features of the biosynthesized bimetallic EC-Ag/MnNPs were analysed using UV-Vis spectroscopy. Two distinct absorption peaks were observed in the UV region at 262 nm and 412 nm (Fig. 1A), corresponding to interband transitions. A slight shift in the absorption spectrum of EC-Ag/MnNPs suggests possible interactions between silver and manganese nanoparticles within the bimetallic system, potentially contributing to their modified optical behaviour. The coexistence of both metals may lead to variations in absorbance values compared to individual nanoparticles (Sherpa et al., 2024; Moeen et al., 2022). The surface plasmon resonance (SPR) band typically associated with silver nanoparticles may be suppressed or shifted due to changes in the electronic environment induced by the presence of manganese. Notably, the presence of silver contributed to an absorbance shift toward the visible region. The optical bandgap of biogenic EC-Ag/MnNPs, computed from Tau’s plot, was found to be 3.34 eV (Fig. 1B), indicating pronounced quantum confinement effects resulting from their reduced size and altered electronic structure compared to bulk silver and manganese (Das et al., 2025; Warsi et al., 2020). The observed bandgap facilitates the efficient absorption of high-energy photons, rendering the EC-Ag/MnNPs suitable for UV-responsive applications such as photocatalysis, optical sensing, and antibacterial activity via reactive oxygen species (ROS) generation (Vikal et al., 2023). The green synthesis strategy not only promotes environmental sustainability but also enhances the biocompatibility of the nanoparticles, thereby expanding their applicability in biomedical and environmental domains. However, the stability of the nanoparticle suspension plays a crucial role in enhancing thermal conductivity. Zeta potential analysis was performed to assess the stability and surface charge of the biosynthesized nanoparticles, revealing a value of approximately −32.1 ± 0.2 (Fig. 1C). The strong negative surface charge effectively prevents nanoparticle aggregation, further supporting their stability in suspension.

Figure 1 Characterization of biosynthesized bimeteallic EC-Ag/MnNPs by Euphorbia cactus extract: (A) UV-vis absorption spectra, (B) Tauc plot, (C) zeta potential analysis, (D) XRD pattern.

The structure, crystallinity and phase purity of the bimetallic EC-Ag/MnNPs was verified through XRD analysis. The sharp and well-defined diffraction peaks observed in XRD pattern (Fig. 1D), indicated the crystalline nature of the biosynthesized nanomaterial. The diffraction peaks indexed to (111), (200), (220), and (311) correspond to cubic phase of silver (JCPDS card no. 04-0783), while the peaks indexed to (222), (431), (440), and (220) were characteristic of the orthorhombic phase of manganese dioxide (JCPDS card no. 86-2337).The XRD spectrum of the bimetallic EC-Ag/MnNPs displayed characteristic peaks corresponding to both silver and manganese, confirming the presence of both phases within the nanoparticles (Tien et al., 2020). These peak positions exactly matched standard reference values, confirming the formation of bimetallic nanoparticle. The formation of a heterostructured crystalline framework was supported by additional peaks, shifts in peak positions, and variations in peak intensities compared to the individual AgNP and MnNP patterns (Ali et al., 2023; Ramesh & Rajendran, 2024), suggesting possible alloying or interactions between the two metals. Some unassigned peaks (denoted by*) were also observed alongside the Ag and Mn crystalline peaks. These may originate because of crystalline organic compounds that were induced during nanoparticle synthesis mediated by Euphorbia cactus plant extract. These findings confirm the formation of nanocrystalline EC-Ag/MnNPs. The average crystallite size of EC-Ag/MnNPs was estimated using the Scherrer equation (D = kλ/βCosΘ) and found to be 16.42 nm, which aligns with previously reported values (Sher et al., 2022).

Figure 2 FTIR spectra of (A) Euphorbia cactus extract and (B) biosynthesized bimeteallic EC-Ag/MnNPs.

The FTIR spectra of the Euphorbia cactus extract and bimetallic EC-Ag/MnNPs revealed important functional moieties and potential chemical interactions involved in the nanoparticle synthesis process (Fig. 2). The FTIR spectrum of the Euphorbia cactus extract exhibited distinct absorption bands corresponding to various functional groups (Fig. 2A). A broad absorption peak at 3,426 cm−1 designated O-H stretching vibrations of polyphenol and –OH groups of sugar ring detected in the Euphorbia cactus extract. Two peaks emerged at 2,924 cm−1 and 2,856 cm−1 were attributed to C-H stretching vibrations of methyl (CH3) and the symmetric C-H stretching vibrations of methylene (CH2) groups, respectively (Binish et al., 2023). A prominent peak observed at 1,607 cm−1, corresponds to C=O stretching vibrations, indicative of different aromatic constituents such as phenols and flavonoids, found in Euphorbia cactus aqueous extract (Bekele & Haile, 2020). Peaks detected at 1,384 cm−1 and 1,276 cm−1 were assigned to medium C-H bending and strong C-O stretching vibrations of ester groups, respectively. A peak noticed around 1,056 cm−1 was ascribed to C-O stretching of primary alcohols, while the band located at 1,632 cm−1 was attributed to O-H bending vibrations of water molecule (Jiang, Koizumi & Yamada, 2000). These spectral profiles indicated that the Euphorbia cactus aqueous extract predominantly contained triterpenoids, flavonoid, phenolics, and steroids components, which served as reducers during the synthesis of nanoparticle. While proteins containing carbonyl, carboxyl, and amine functional moieties can assist in stabilize the nanoparticles by attaching to their surfaces. However, other phytocomponents including reducing sugars, carboxylic acids, and amines may contribute to electrons donations. In contrast, the FTIR spectrum of pre-synthesized EC-Ag/MnNPs (Fig. 2B), showed distinctive peaks at 1,626 cm−1, 3,440 cm−1 and 605 cm−1, which correspond to C–H stretching vibrations, O-H functional moieties of polyphenolic compounds, and Mn-O bending vibrations, respectively (Fang & Compton, 1988). Notable shifts in the intensities and positions of peaks related to hydroxyl, carbonyl, and methyl groups between the extract and nanoparticles indicate the involvement of phytochemicals in both the reduction of Ag+ and Mn2+ ions and the stabilization of the bimetallic EC-Ag/MnNPs. These spectral changes confirm the dual role of Euphorbia cactus phytoconstituents as reducing and capping agents in nanoparticle formation (Deshpande, Gothalwal & Chandra, 2023).

Figure 3 (A) SEM image at × 50,000 magnification, (B) EDX spectrum, (C) TEM image × 80,000 magnification, (D) particle size distribution histogram, and (E) SAED pattern of biosynthesized bimeteallic EC-Ag/MnNPs using Euphorbia cactus extract.

The surface topography and morphological features of the bimetallic EC-Ag/MnNPs synthesized from Euphorbia cactus extract was investigated by SEM images. As shown in Fig. 3A, the nanoparticles exhibited a three-dimensional, porous architecture with randomly oriented and interconnected structures. The formation of relatively uniform, spherical-shaped particles was distinctly visible, indicating homogeneity in morphology. SEM image also revealed some agglomeration and adhesion among the nanoparticles, attributed to polarity and electrostatic interactions of MnNPs during green synthesis, as well as their nanoscale dimensions (Ahmad, Yaqoob & Gul, 2022). The addition of Ag+ ions to the MnNP surface did not significantly alter the morphology, though grain growth was evident. Phytochemicals present in the plant extract facilitated the formation of mesocrystals through a self-assembly process, playing a critical role in crystal growth and structural development (Zaragosa et al., 2024). The elemental content and relative abundance of biogenic EC-Ag/MnNPs was assessed by EDX analysis. The horizontal axis represents energy (KeV), while the vertical axis indicates X-ray counts. EDX spectrographic analysis of bimetallic EC-Ag/MnNPs alloy revealed prominent signals corresponding to silver (Ag) at 43.62%, oxygen (O) at 27.54%, manganese (Mn) at 18.71% and carbon (C) at 10.13% (Fig. 3B). Characteristic emission peaks for Ag and Mn, confirmed their successful integration into the bimetallic nanoparticles. Overall, EDX analysis validated the elemental purity and chemical composition of the biogenic EC-Ag/MnNPs, with minor carbon signals attributed to organic capping agents derived from the Euphorbia cactus extract. Further, morphological assessment using TEM (Fig. 3C) and corresponding particle size distribution histograms (Fig. 3D), showed well-dispersed, predominantly spherical nanoparticles ranging from 12.214 to 18.327 nm, with an average size centred around 16.23 nm. Additional structural characterization of EC-Ag/MnNPs using SAED (Fig. 3E) displayed distinct diffraction rings attributed to the orthorhombic Mn phase (light yellow rings) and face-centred cubic Ag phase (blue rings). TEM visuals revealed (Fig. 3C) an interplanner spacing of 0.267 nm associated with the (222) plane of orthorhombic Mn, indicating preferential growth along this plane (Swaminathan et al., 2024). A lattice spacing of 0.232 nm, attributed to the (111) plane of cubic Ag, was also observed (Song et al., 2024). These results are consistent with the XRD findings presented in Fig. 1D.

Antimicrobial activity

The bioactive potential of EC-extract and biogenic bimetallic EC-Ag/MnNPs was evaluated against bacterial strains including Staphylococcus aureus, Staphylococcus pneumonia, MRS, Enterobacter cloacae, Escherichia coli, and Salmonella Typhi, as well as fungal strains C. glabrata and C. parapsilosis. The antimicrobial activity of EC-extract and EC-Ag/MnNPs against the aforementioned microbial strains at varying concentrations was presented in Table S1 of the supporting information. Antimicrobial potency was indicated by the size of the inhibition zones surrounding the wells in each Petri dish. The results clearly indicate that the zone of inhibition (ZOI) increases proportionally with the concentration of the test samples. The antibacterial efficacy EC-extract and EC-Ag/MnNPs was assessed using the well diffusion experiment and presented in Fig. 4. The results showed that biosynthesized EC-Ag/MnNPs exerted markedly superior antibacterial effectiveness towards all tested microorganisms in comparison to EC-extract. In most cases, EC-Ag/MnNPs produced significantly larger zones of inhibition (ZOIs), although the extract displayed relatively stronger activity against a few strains. Both EC-extract and EC-Ag/MnNPs were effective towards Gram-negative and Gram-positive test strains. At 100 µg/mL concentration, EC-Ag/MnNPs exhibited pronounced bactericidal effects against Gram-negative Escherichia coli (38.15 ± 0.32 mm) and Salmonella Typhi (36.81 ± 0.51 mm) bacterial strains, whereas the EC-extract produced moderate inhibition zones of 18.36 ± 0.09 mm and 17.42 ± 0.21 mm, respectively, at the same dose. Among the Gram-positive strains, MRSA showed the highest sensitivity with ZOIs of 25.36 ± 0.09 mm for EC-Ag/MnNPs and 21.14 ± 0.04 mm for the EC-extract. The complete ZOI values for the tested bacterial strains were summarized in Table 2. However, antifungal testing revealed that C. glabrata was the most susceptible fungal strain, with ZOIs of 35.10 ± 0.39 mm for the pre-synthesized bimetallic EC-Ag/MnNPs and 33.82 ± 0.97 mm for the EC-extract (Fig. 4, Table 2). Based on antimicrobial results, three most susceptible strains, Escherichia coli, Salmonella Typhi, and C. glabrata were selected for MIC and MBC estimation. The corresponding MIC and MBC values for the EC-extract and biogenic bimetallic EC-Ag/MnNPs were presented Table 3. Bimetallic EC-Ag/MnNPs have demonstrated enhanced bactericidal activity compared to EC-extract against all tested bacterial strains, owing to their intrinsic antibacterial properties. The bactericidal effect of EC-Ag/MnNPs nanoparticles surpasses that of the antibiotic Gentamycin. Although the precise antimicrobial mechanism of metal and metal oxide nanomaterials remains not fully understood, it is widely accepted that they act through multiple pathways, including the generation of ROS, disruption of the bacterial cell wall through direct contact, and inhibition of protein synthesis, ultimately resulting in cell death (Parvin, Joo & Mandal, 2025; Godoy-Gallardo et al., 2021). The antimicrobial effectiveness of hybrid nanoparticles is influenced by factors including particle size, the nature of capping agents, and the synthesis method employed. Smaller nanoparticles, with a higher surface-to-volume ratio, exhibit enhanced antimicrobial activity (Dubey et al., 2024; Rai et al., 2024). However, plant phytoconstituents act as natural capping agents, imparting a more negative surface charge to the nanoparticles by binding to them. This increased negative charge contributes to nanoparticle stability and augments their bactericidal potential (Villagrán et al., 2024; D’Souza et al., 2024). In this study, the disc diffusion method was utilized to access the antibacterial activity, leveraging the efficient diffusion properties of silver (Ag) and manganese (Mn) ions through agar matrices. The biosynthesised bimetallic EC-Ag/MnNPs readily release these ions upon contact with bacterial cells, enhancing their bactericidal effect. Silver ions (Ag+) are known to disrupt membrane integrity, interfere with vital cellular processes, and inhibit bacterial proliferation (Thurman, Gerba & Bitton, 1989), while manganese (Mn2+) further contribute by interfering with enzymatic functions, altering metabolic pathways, and potentially inducing oxidative stress (Culotta & Daly, 2013). The concurrent release of both ions produces a synergistic effect that amplifies the antibacterial efficacy of EC-Ag/MnNPs. Moreover, the nano-sized (18.32 nm), spherical structure of bimetallic EC-Ag/MnNPs offers a large surface area, facilitating enhanced ion release and greater interaction with bacterial cell membranes (Bahrami, Delshadi & Jafari, 2020). This increased surface area also promotes the ROS generation, which can damage essential biomolecules such as proteins, lipids, and DNA (Bahrami, Delshadi & Jafari, 2020). The strong antibacterial activity of small-sized nanoparticles has been validated by numerous studies (Modi et al., 2023; Ji et al., 2020). A major factor in antibiotic resistance among many pathogens is the limited membrane permeability of hydrophilic antibiotics, restricting their intracellular accumulation. In contrast, nanoparticles can overcome this barrier, especially in phagocytic cells where adsorbed NPs exhibit enhanced intracellular activity (Wu et al., 2022). The proposed mechanism of nanoparticle-induced antibacterial action is illustrated in Fig. 5.

Figure 4 Well diffusion assay demonstrating the antibacterial activity of Euphorbia cactus extract and biosynthesized bimeteallic EC-Ag/MnNPs against eight different microbial strains.

Table 2 Zone of inhibition (mm) of EC-extract and biosynthesized EC-Ag/MnNPs against various microorganisms.

	Microbial strains	Zone of inhibition (mm)	
		EC-extract	EC-Ag/MnNPs	Standard	
	Staphylococcus aureus	16.08 ± 0.1	22.13 ± 0.17	20.16 ± 0.4	
Gram +ive	Staphylococcus pneumonia	7.26 ± 0.91	5.48 ± 0.64	18.23 ± 0.37	
	MRSA	21.14 ± 0.04	25.36 ± 0.09	22.12 ± 0.32	
Gram -ive	Enterobacter cloacae	8.32 ± 0.97	24.71 ± 0.04	19.26 ± 0.51	
Escherichia coli	18.36 ± 0.09	38.15 ± 0.32	20.17 ± 0.35	
Salmonella Typhi	17.42 ± 0.21	36.81 ± 0.51	20.23 ± 0.38	
Fungi	C. glabrata	25.14 ± 0.40	35.10 ± 0.39	19.17 ± 0.50	
C. parapsilosis	27.73 ± 0.38	33.82 ± 0.97	20.21 ± 0.94	
Notes.

± The experimental results in triplicate.

Table 3 Values of MIC and MBC for the EC-extract and green synthesized EC-Ag/MnNPs.

Microbial strains	EC-extract	EC-Ag/MnNPs	
	MIC mg /ml	MBC mg/ml	MIC µg /ml	MBC µg/ml	
Salmonella Typhi	4.23	13.21	64.12	94.12	
Escherichia coli	3.67	10.45	48.42	86.12	
C. glabrata	22.72	50.34	85.46	180.21	

Figure 5 Schematic representation of the probable mechanism of antimicrobial activity of biosynthesized EC-Ag/MnNPs.

It has been observed that EC-Ag/MnNPs attached cell surface of bacteria and alter cell membrane potential, induce ROS product.

Notably, EC-Ag/MnNPs exhibited stronger inhibitory effect against Gram-negative bacteria than Gram-positive strains. This disparity in susceptibility may arise from variations in bacterial physiology, metabolic activity, cell wall structure, and extent of nanoparticle interaction. The penetration efficiency of EC-Ag/MnNPs is influenced by both the structural composition of the bacterial cell wall and membrane, as well as physicochemical attributes of the nanoparticles (Ahmed et al., 2020). Gram-positive bacteria possess a thick, rigid peptidoglycan layer reinforced with peptide chains, while Gram-negative bacteria have a comparatively thinner coating (Silhavy, Kahne & Walker, 2010). This thinner peptidoglycan layer in Gram-negative bacteria enables easier access of nanoparticles to intracellular targets, thereby inhibiting cell regeneration (Ahmed et al., 2018). Moreover, the negatively charged outer membrane of Gram-negative bacteria facilitates electrostatic interaction with positively charged nanoparticles, thereby enhancing their cellular uptake (Haffner & Malmsten, 2017).

Furthermore, out of eight tested microbial strains, the two most sensitive strains (Escherichia coli and Salmonella Typhi) were selected for scanning electron microscopy (SEM) analysis. SEM images revealed distinct morphological alterations in Escherichia coli and Salmonella Typhi cells treated with EC-extract and EC-Ag/MnNPs, compared to untreated cells. Untreated cells appeared intact, smooth and regularly shaped, while treated cells showed visible structural damage (Fig. 6). Cells exposed to the EC-extract exhibited mild membrane rupture (Figs. 6B and 6E), whereas those treated with biogenic EC-Ag/MnNPs displayed extensive cellular disruption, including ruptured and lysed cells (Figs. 6C and 6F). These observations indicated that EC-Ag/MnNPs compromise both cell morphology and potentially DNA integrity. However, this effect may be a secondary, as the cytoplasmic membrane is generally the primary target during nanoparticle interaction. SEM examination confirmed that Ag+ and Mn2+ ions released from EC-Ag/MnNPs disrupt the cell membrane before penetrating the cytoplasm and interacting with cellular components (Kedziora et al., 2018). These findings support the notion that EC-Ag/MnNPs function as multi-target antibacterial agent, primarily acting at the cytoplasmic membrane while also affecting other vital cellular structures.

Figure 6 SEM images of (A, D) untreated Escherichia coli and Salmonella Typhi with intact cells, (B, E) EC-extract treated, and (C, F) EC-Ag/MnNPs treated Escherichia coli and Salmonella Typhi, showing morphological changes in the cells.

The antimicrobial potential of biosynthesized bimetallic EC-Ag/MnNPs using Euphorbia cactus extract was compared with previously reported green-synthesized bimetallic Ag/MnNPs using different plant extracts, as presented in Table 4. The results clearly demonstrate that biogenic bimetallic EC-Ag/MnNPs exhibit stronger antibacterial and antifungal potential than those reported in earlier studies.

Table 4 Some previous studies conducted on plant-mediated synthesis of bimetallic Ag/MnNPs using different plant extracts and their antimicrobial potential.

Plant name	Plant used	Nanoparticle	Antimicrobial activity	Zone of inhibition	References	
Cordia macleodii	Leaves	Ag-MnNPs	Antibacterial	≥15 mm, ≥12 mm	Deshpande, Gothalwal & Chandra (2023)	
Hibiscus rosa-sinensis	Leaves	Ag-MnONPs	Antifungal	≥11.0 mm, ≥10.0 mm, ≥ 7.0 mm	Roy et al. (2025)	
Leucophyllum frutescens	Leaves	AgNPs, MnONPs	Antibacterial	≥15.1 mm, ≥11.6 mm	Al-Masoud et al. (2024)	
Arachispintoia	Leaves and stem	Ag-MnNPs	Antibacterial	≥19.7 mm, ≥17.7 mm	Tien et al. (2020)	
Cucurbita pepo	Leaves	AgNPs, MnO2NPs, Ag-doped MnO2NPs	Antibacterial	≥14.0 mm, ≥13.0 mm, ≥ 15.0 mm	Krishnaraj et al. (2016)	

Antioxidant activity

The free radical scavenging potential of biosynthesized bimetallic EC-Ag/MnNPs, aqueous EC-extract and L-ascorbic acid were assessed by the antioxidant DPPH, with the results illustrated in Fig. 7 and corresponding values detailed in Table 5. To ascertain their relative efficacy, tests were performed at four varied concentrations (20, 40, 60 and 80 µg/mL). The DPPH scavenging ability of EC-Ag/MnNPs, aqueous EC-extract, and L-ascorbic ranged from 28.32 to 96.12%, 21.34 to 75.16, and 25.45 to 87.34%, respectively. Absorbance of the reaction mixture (test sample with DPPH methanol solution) was recorded at 517 nm, where lower absorbance value indicated higher free radical scavenging potential. The findings suggested that EC-Ag/MnNPs have a greater ability to scavenge free radicals, which increases proportionally with the dose. At a higher concentration (80 µg/mL), biogenic EC-Ag/MnNPs demonstrated superior free radical scavenging efficiency compared to both EC-extract and ascorbic acid. The phytochemical components associated with EC-Ag/MnNPs exhibited the highest scavenging activity, reaching 96.12% at 80 µg/mL. The antioxidant effectiveness was assessed by calculating the IC50 value using linear regression analysis of the inhibition rate (%) versus the concentration of the test samples. The bimetallic EC-Ag/MnNPs exhibited the lowest IC50 value (42 µg/mL), indicating superior antioxidant efficiency compared to the EC-extract (58 µg/mL) and standard ascorbic acid (48 µg/mL). This lower IC50 value reflects the stronger DPPH free radical scavenging ability of EC-Ag/MnNPs, consistent with previous studies (Alam et al., 2022).

Figure 7 In vitro free radical scavenging activity of EC-Ag/MnNPs, EC-extract, and ascorbic acid at different concentrations.

Table 5 In vitro DPPH scavenging activities of EC-extract, biosynthesized EC-Ag/MnNPs and ascorbic acid.

Samples	Inhibition (%)	
	Concentration (µg/mL)	
	20	40	60	80	IC50 value	
EC-extract	21.34 ± 0.52	38.12 ± 1.38	58.63 ± 0.49	75.16 ± 0.04	58.12	
EC-Ag/MnNPs	28.32 ± 0.80	47.96 ± 0.61	75.13 ± 0.03	96.12 ± 0.40	42.89	
L-Ascorbic acid	25.45 ± 0.08	42.15 ± 1.05	69.27 ± 0.46	87.34 ± 0.93	48.74	
Notes.

± showed the results of the triplicate experiment.

Ecological safety of EC-Ag/MnNPs

The green synthesis of EC-Ag/MnNPs offers a sustainable and ecologically safe alternative to conventional chemical synthesis methods. Utilizing EC-extract acts as a natural reducing agent for the fabrication of bimetallic EC-Ag/MnNPs, this eco-friendly approach eliminates the use of toxic solvents, harsh reducing agents, and hazardous stabilizers, significantly reducing environmental burden. Unlike chemically synthesized nanoparticles, green-synthesized EC-Ag/MnNPs exhibit reduced toxicity toward non-target organisms, including soil microbes, aquatic species, and plants. The use of natural capping and stabilizing agents enhances the biocompatibility and degradability of the nanoparticles, minimizing long-term accumulation in ecosystems. Furthermore, the biosynthetic route enables better control over nanoparticle size, shape, and surface functionality, which can be tailored to reduce undesirable interactions with living systems. Studies have shown that green-synthesized monometallic or bimetallic nanoparticles degrade more readily under environmental conditions and exhibit lower bioavailability, decreasing the risk of bioaccumulation and tropic transfer in food chains (Larranaga-Tapia et al., 2024; Abuzeid et al., 2023). Overall, the green synthesis of EC-Ag/MnNPs aligns with principles of green chemistry and sustainable nanotechnology, offering promising applications in agriculture, medicine, and environmental remediation with minimal ecological risks.

Conclusion

The study established a sustainable, simple, and efficient green synthesis of bimetallic EC-Ag/MnNPs using phytoconstituent-rich extract from aerial parts Euphorbia cactus as a mediating agent. Structural and optical characterisation confirmed the successful formation of EC-Ag/MnNPs, demonstrating that the nanoparticles were pure and composed of excessively nano-sized Ag and MnNPs particles. SEM and TEM analyses revealed well-dispersed, spherical EC-Ag/MnNPs with an average diameter of 18.32 nm, while XRD patterns validated their crystalline nature. Both biosynthesized EC-Ag/MnNPs and EC-extract exerted significant antimicrobial and antioxidant properties; however, the bimetallic EC-Ag/MnNPs showed more potent antimicrobial properties against all investigated microbial strains, with Escherichia coli and Salmonella Typhi being the most sensitive strains. Additionally, bimetallic EC-Ag/MnNPs displayed superior antioxidant capability compared to both the extract and the standard. In conclusion, this study underscores the potential of bimetallic EC-Ag/MnNPs synthesized using Euphorbia cactus extract as effective bioactive agents. These hybrid nanomaterials offer a promising alternative to conventional therapies for combating multidrug-resistant pathogens. Nonetheless, further investigations are required to elucidate the underlying biochemical mechanisms responsible for their antibacterial and antioxidant activities. Future research should focus on optimizing synthesis conditions to enhance their efficacy and expand their application in in vivo toxicity studies, as well as biomedical and environmental fields, contributing to the development of next-generation functional nanomaterials.

Supplemental Information

Supplemental Information 1 Antibacterial activity of EC-extract and biosynthesized EC-Ag/MnNPs against different bacterial and fungal strains at different concentrations

Supplemental Information 2 Raw data of Zone of Inhibition (mm) of EC-extract and biosynthesized EC-Ag/MnNPs against various microorganisms

Supplemental Information 3 Raw data of antioxidant activity of EC-extract, biosynthesized EC-Ag/MnNPs and ascorbic acid

The authors gratefully acknowledge the Scientific Research Unit, Inaya Medical Colleges, Saudi Arabia for allowing the use of instrumentation: specifically, the scanning electron microscope (SEM), transmission electron microscope (TEM), and selected area electron diffraction (SAED) analyses of the prepared bimetallic nanoparticles (EC-Ag/MnNPs), which were conducted at the Scientific Research Unit, Inaya Medical Colleges, Saudi Arabia.

Additional Information and Declarations

Competing Interests

Author Contributions

Data Availability

The authors declare there are no competing interests.

Gadah A. Al-Hamoud conceived and designed the experiments, authored or reviewed drafts of the article, and approved the final draft.

Musarat Amina performed the experiments, prepared figures and/or tables, authored or reviewed drafts of the article, and approved the final draft.

Nawal M. Al-Musayeib analyzed the data, authored or reviewed drafts of the article, and approved the final draft.

Samiah Alhabardi conceived and designed the experiments, prepared figures and/or tables, authored or reviewed drafts of the article, and approved the final draft.

Mohsin Ul Haq conceived and designed the experiments, prepared figures and/or tables, and approved the final draft.

Saeed Akhtar performed the experiments, analyzed the data, authored or reviewed drafts of the article, and approved the final draft.

The following information was supplied regarding data availability:

The raw data is available in the Supplemental Files.

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
