# Peer review of "Antimicrobial and scavenging potential of green synthesized silver/manganese bimetallic nanoparticles using Euphorbia cactus extract"

_PeerJ, doi:10.7717/peerj.20244_

## Round 0.1 · original submission · Major Revisions

Kindly respond to each of the reviewers' comments in a detailed rebuttal letter.

**Language Note:** The review process has identified that the English language must be improved. PeerJ can provide language editing services - please contact us at [email protected] for pricing (be sure to provide your manuscript number and title). Alternatively, you should make your own arrangements to improve the language quality and provide details in your response letter. – PeerJ Staff

Reviewer 1 ·

Basic reporting

The manuscript "Antimicrobial and Scavenging Potential of Green Synthesized Silver/Manganese Bimetallic Nanoparticles Using Euphorbia Cactus Extract" is well written based on clarity, scientific rigor, methodology, and structure. However, I recommend Major revision before acceptance.
1. The manuscript needs minor grammatical refinements for better readability. For example, phrases like "The formed EC-Ag/MnNPs were observed to be well-dispersed and spherical" can be simplified to "The EC-Ag/MnNPs were well-dispersed and spherical."
2. The introduction effectively outlines the problem but should incorporate more recent studies on the toxicity and biomedical applications of bimetallic nanoparticles.Include recent studies: https://doi.org/10.1080/20550324.2024.2414137, https://doi.org/10.1007/s13369-024-08968-w, https://doi.org/10.1007/s11356-022-23500-z

3. Figures and Tables: Figure legends should be more detailed, particularly in describing key observations in characterization images.
4. Reference Formatting: Some references (e.g., Franco et al., 2023) are missing DOIs or web links.
5. Ensure consistency in abbreviations (e.g., use either EC-Ag/MnNPs or Ag/MnNPs throughout the text).
6. Clearly state why Ag/MnNPs were selected over other potential bimetallic combinations. Give a table on enhancement of bioactivity with this bimetallic combinations, supported with literature survey
7. Provide more details on reaction conditions such as pH, stirring speed, and precise temperature control.
8. The crystallite size calculated using the Scherrer equation needs to be validated with additional size distribution data from TEM.

Experimental design

no comment

Validity of the findings

9. Identify the specific functional groups from the E. cactus extract that contribute to nanoparticle stabilization.
10. The study should include a negative control (e.g., untreated bacteria/fungi) to establish baseline microbial growth.
11. Provide statistical significance (p-values) for antimicrobial and antioxidant results.
12. Discuss how the antimicrobial efficacy of EC-Ag/MnNPs compares with conventional antibiotics.
13. Explain why EC-Ag/MnNPs exhibited higher efficacy against Gram-negative bacteria.
14. The authors should elaborate on the possible interaction mechanisms between Ag/MnNPs and microbial cell membranes.
15. Provide more details on sample preparation for repeatability, including the number of replicates used per experiment.

Additional comments

General Comments
1. Avoid generic statements like "this study highlights the potential of EC-Ag/MnNPs..."—instead, summarize the key novelty and impact.
2. The discussion should include a short section on the ecological safety of EC-Ag/MnNPs.
3. Expand on how these nanoparticles could be practically applied in medical or industrial settings.
4. The authors should suggest further areas of research, such as in vivo toxicity studies or biomedical applications.

Reviewer 2 ·

Basic reporting

1.Tauc’s plot image is missing
2. information about phase of bimetallic nanoparticles through XRD is not shown
Is it core shell, or mixed phase?

Experimental design

1 Which part of the plant material was used in this study? If whole plant? Then Why?
2. What is the exact ratio of Metal ion concentration and plant extract?
3. In antimicrobial activity, only volumes of samples are reported not concentration. (concentration dependent activity must be reported also in result).
4. No clear explanation to UV-Vis spectroscopy analysis is given Discussion with the previous studies is missing.

Validity of the findings

Report of the previous studies carried out on synthesis of various nanomaterials using Euphorbia cactus is missing. The literature need to be improved

Additional comments

1.The Paragraph -3 of Introduction is to be improved and need to be more scientifically.
2. In the SEM image no homogeneous particle shapes are shown
3. SAED pattern in TEM analysis is not found
4.The quality of all figures are to be improved. May be using Photoshop tools the figure panels will give clear images
5.Place full images of Figure 3.
6.Requires high quality scale bar on TEM micrograph

Reviewer 3 ·

Basic reporting

Reviewers report
ID manuscript: 114348-v0
Title: Antimicrobial and scavenging potential of green synthesized silver/manganese bimetallic nanoparticles using Euphorbia cactus extract
Comments and observations:
Several sentences in the manuscript need to be improved by the authors. They have been pointed out (yellow color) in the manuscript.
Abstract:
Some parts of written sentences should be placed more directly to improve the understanding of the concepts that the authors wish to express, which have been pointed out in yellow in the manuscript.
In this section the general information of the manuscript is presented. However, there is no clarity and clear structure of the section. There is no mention of how the synthesis of the Ag/Mn nanoparticles was carried out. The process of obtaining the Euphorbia cactus extract is not mentioned either.
Introduction
The first observation of this section is its length. It is a very long section, a reduction is recommended and the most relevant and important information of the state of the art of the research should be presented. Where it is pointed out what has been reported in the literature.
The literature review is required to be more timely and accurate, being the most important information to be presented in this section.
On the other hand, the last paragraph of this section (Introduction) is very important. In this paragraph, the procedures and techniques used can be presented, as well as an application of the product that has been developed.
There are several errors in spelling, wording, incorrect use of words, lack of articles and prepositions in sentences, words without a blank between them, among others. A revision of this section is recommended and by a reviewer whose native language is English.

Experimental design

In the Materials and Methods section, a large number of spelling and writing errors, incorrect use of prepositions, lack of articles, incorrect use of abbreviations of the international system can be observed. These and other errors have been pointed out (highlighted) in the document. It is suggested that in this section a homogeneity of terms and units be made.
In the section Biogenic synthesis of bimetallic EC-Ag/MnNPs, it is important that the authors clearly describe the concentrations of the solutions and the amount of the reagents used in the synthesis of the nanoparticles.

Validity of the findings

On the other hand, the Results and Discussion section is very general in some parts. It was expected that the subsections (where these are not identified) would be approached and analyzed from a more in-depth and analytical point of view. In addition, the authors do not discuss some subsections; for example, the subsection between lines 298-317. Most of the discussion is very superficial and not deep. It s very important that the authors must revise the mistakes highlighted in the manuscript.

Conclusion section. I strongly suggest that the author revise and re-write some statements. Several sentences in this section are very general (observations) and do not provide a conclusion. My suggestion is: The conclusion section provides an objective analysis of the results the evidence from the literature supports the statements.

Additional comments

Although AI is a tool for editing information in a review, it does not reach the capacity of human reasoning to analyze the information. It is recommended that authors revise the manuscript and limit the use of AI to manuscript refinement and spell-checking. A native English speaker with knowledge and experience in this field must revise this manuscript.
In the References section, it can be observed that several references are not spelled correctly; they have been marked in yellow color.
I have enclosed your manuscript revised with the points to be revised (these are highlighted).

Annotated reviews are not available for download in order to protect the identity of reviewers who chose to remain anonymous.

---

## Round 0.2 · Minor Revisions

Please address the reviewers' comments and submit a revised version.

Reviewer 3 ·

Basic reporting

Second review
ID manuscript: 114348-v0
Title: Antimicrobial and scavenging potential of green synthesized silver/manganese bimetallic nanoparticles using Euphorbia cactus extract
Comments and observations:
The authors have made the modifications suggested in the first review. There is a substantial improvement in the wording of the sentences and in the use of technical and scientific language required for an article in this journal.
However, we have the following observations regarding this revised manuscript:
1) It can be seen that several words are not separated, i.e., there is no space between them. These have been clearly marked in yellow in the manuscript. For example, “E.cactus” instead of “E. cactus” The words with these markings are found on lines: 20, 26, 28, 78, 114, 117, 118, 126, 195, 238, 239, 354, 357, 403, 483, and 494.
2) Authors are advised to make changes to physical units of time (hours) in accordance with the International System of Units. Generally, “hours” is abbreviated as “h”, “milliliters” as “mL”.
3) In line 420 “Tables” should be “Table”.

Experimental design

The following comments are made regarding the experimental section:
1) The experimental procedures used by the authors are well described and clear enough for readers to reproduce them.
2) Clarifications have been made regarding the correct abbreviations to be used for physical units.
3) The methods described include references that support the procedures used by the authors.

Validity of the findings

It can be seen that this manuscript is relevant from the point of view of a method for synthesizing silver/manganese nanoparticles using natural extracts. The yield values of the synthesis procedure are very interesting and comparable to other procedures that cannot be considered green chemistry.
As mentioned above, the authors have made substantial modifications to the manuscript and improved the wording of some sections, such as the conclusions, which allow the research objectives reported in this manuscript to be addressed.
The main findings highlighted in the conclusions are supported by the results presented by the authors

Additional comments

Authors are advised to make the corrections indicated in section 1.

Annotated reviews are not available for download in order to protect the identity of reviewers who chose to remain anonymous.

---

## Round 0.3 · accepted · Accept

Congratulations! Your paper has been accepted in PeerJ